# Interpreting fMRI Studies in Populations with Cerebrovascular Risk: The Use of a Subject-Specific Hemodynamic Response Function

**DOI:** 10.3390/bs15111457

**Published:** 2025-10-26

**Authors:** Ian M. McDonough, Andrew R. Bender, Lawrence Patihis, Elizabeth A. Stinson, Sarah K. Letang, William S. Miller

**Affiliations:** 1Department of Psychology, Binghamton University, Binghamton, NY 13902, USA; 2Cleveland Clinic, Lou Ruvo Center for Brain Health, Las Vegas, NV 89106, USA; 3Department of Psychology, University of Portsmouth, Portsmouth PO1 2DY, UK; lawrence.patihis@port.ac.uk; 4Department of Psychiatry and Behavioral Sciences, Emory University School of Medicine, Atlanta, GA 30322, USA; elizabeth.ashley.stinson@emory.edu; 5Department of Neurology, University of Alabama, Birmingham, AL 35233, USA; skletang@uab.edu; 6Department of Emergency Medicine, University of Alabama, Birmingham, AL 35233, USA; wsmiller@uabmc.edu

**Keywords:** aging, hemodynamic response function, functional magnetic resonance imaging, encoding, episodic memory, vascular risk

## Abstract

Functional magnetic resonance imaging (fMRI) is commonly used to investigate the neural bases of aging and psychological disorders. However, the BOLD signal captured by fMRI is affected by many factors that are non-neural in origin. We tested how vascular health risks, which often go unmeasured in neuroimaging studies, and aging interact to modify the shape and/or timing of the HRF, which then affect the differences in patterns of brain activity in a task-evoked memory encoding paradigm. Adult participants (aged 20–74) answered questions about their health and underwent two fMRI tasks: viewing a flashing checkerboard and a memory encoding task. Aging and vascular risk had the largest impacts on the maximum peak value of the HRF. Using a subject-specific HRF resulted in a dampening of brain activity in task-positive and task-negative regions. Across three vascular risk factors, using a subject-specific HRF resulted in more consistent brain regions that reached significance and larger effect sizes compared with the canonical HRF. These findings serve as a cautious tail when interpreting task-evoked fMRI activity, especially in populations experiencing alterations to brain vasculature including many older adults and people with neurocognitive disorders like Alzheimer’s disease and related dementias.

## 1. Introduction

For over three decades, functional magnetic resonance imaging (fMRI) has been a primary method for investigating neural function both under optimal conditions and in the presence of alterations due to development, aging, and disease. Nevertheless, fMRI only indirectly measures neural activity by utilizing blood oxygen level dependent (BOLD) responses ([12]; [47]). The physiological link between the BOLD signal and neural activity is complex and can be disrupted under conditions that alter cerebrovascular health ([17]; [19]; [42]; [74]; [86]). Differences or changes in cerebrovasculature due to aging or pathology may modify the BOLD response, even in the absence of true changes in neural activity. Such changes in neurovascularization would likely diverge from the stereotypical model of the neurovascular system used to norm traditional fMRI modeling parameters. This divergence would therefore be a potential source of systematic error in fMRI’s proxy-based measurement of neural activity. Nevertheless, many studies investigating aging, health, and disease continue to use fMRI to infer differential brain functioning despite possibly violating the link between the BOLD signal and neural activity. A review by [55] ([55]) reported that 59–78% of aging studies did not have clear subject-selection criteria for two factors known to change cerebrovascular health (i.e., hypertension diagnosis or Type 2 diabetes), and only about 30% of studies acknowledged the influence of cerebrovascular health on their results. These numbers suggest many extant findings may include inflated false positives or false negatives ([83]). There are multiple routes by which aging and declines in cerebrovascular health can modify the link between the BOLD signal and neural activity. The present study examined the impact of age and vascular risk factors on the shape and timing of the BOLD signal in response to a task and how accounting for alterations of the BOLD signal may alter interpretations of task-evoked brain activations using fMRI.

### 1.1. Vascular Changes in Older Adults and Brain Function

Older age is associated with multiple alterations in metabolic function and vascular physiology. These alterations contribute to increased risk for a compromised or altered cerebrovascular system, possibly leading to stroke, myocardial infarction or even dementia ([2]; [14]; [24]; [22]). Although much research has been conducted on these risk factors in older age, many vascular changes begin to occur in midlife and are predictive of poor cognitive and health outcomes (e.g., [38]). Both age and associated vascular changes are linked with increased risk for neurocognitive declines and fMRI techniques have been a prominent method for studying their effects on the brain when engaged in cognitive tasks, including memory (e.g., [5]; [9]; [10]; [16]; [50]; [84]). In a review of the effects of overall vascular risk (e.g., hypertension, heart disease, diabetes, BMI, and smoking) on memory encoding-related brain activity, [53] ([53]) found that such risks were consistently associated with greater break activity in occipital, parietal, and temporal cortex and lower brain activity in the medial temporal lobe (MTL). Specifically, elevations in body mass index and blood pressure in cognitively healthy older adults were associated with widespread patterns of increased activation during a verbal memory encoding task ([10]). Other studies have provided evidence that vascular risk factors are also associated with in an inability to deactivate brain regions with the default mode network (DMN) ([10]; [16]; [50]; [84]). On the surface then, these studies and others have the potential to reveal the neural mechanisms underlying brain health that might put middle-aged and older adults at risk for future cognitive decline.

Both older age and elevated vascular risk are associated with numerous cerebrovascular changes, including altered patterns of BOLD response ([19]; [26]). Such changes in the cerebrovascular system due to age or disease may result in reduced vessel elasticity, increased atherosclerosis, vascular atrophy and degeneration, thickened basement membrane, reduced vessel elasticity, endothelial dysfunction and apoptosis, and amyloid deposition (e.g., [41]). Aging is also associated with microvascular modifications including the loss of cortical capillary networks ([23]), increased vascular morphological complexity and tortuosity, atherosclerosis, and declines in microvessel density, which appear to reduce vascular flow and perfusion ([11]; [30]). Indeed, hypoxia and inadequate blood supply appear to be key mechanisms linking altered age-related vascular dysfunction with cognitive impairment and dementia ([73]). Age-related vascular changes that appear to promote vascular risk are numerous, although the specific and differential contributions of each vascular risk on the BOLD signal are unclear. These factors might reduce resting cerebral blood flow or decrease resting cerebral metabolic rate of oxygen consumption (CMRO2), either of which can significantly alter the BOLD signal ([17]; [19]; [42]; [74]; [86]).

### 1.2. Estimating Task-Evoked Brain Activity in fMRI

The alterations in the BOLD signal mentioned above have important implications for how neural activity is assessed using task-related fMRI. Although much research has noted that age and vascular risk can lead to a decoupling between neuronal activity and the BOLD signal, fewer studies have investigated how these factors influence the shape or timing of the BOLD response to a task. This shape and timing additionally affect how brain activity is estimated. Note that we use the term “BOLD” signal or response to refer to the raw signal from the fMRI scanner, use the term “brain activity” to refer to the estimations of neural activity derived from the BOLD signal, and use the term “neural activity or recruitment” to refer to the hypothetical neural firing that is occurring to which we do not have direct access. Many fMRI software packages rely on a predicted BOLD response, as reflected in the canonical hemodynamic response function (HRF) that is based on a healthy vascular system. Nevertheless, even among healthy young samples, substantial within- and between-subject variability exists ([29]; [56]), suggesting that both healthy aging and age-related vascular pathology are sources of additional variability in the HRF ([19]; [26]). In fact, incorrect estimation of HRF onset time by as little as 1 s can affect statistical sensitivity, resulting in false negatives ([8]; [29]). Therefore, disentangling the effects of age-related neurovascular alterations on BOLD correlates of neural activity from those due to aging or maturational effects in neural recruitment represents a fundamental challenge for fMRI research on the aging brain ([19]).

Previous research has supported the idea that correcting the BOLD signal among older adults can alter the interpretation of the BOLD signal. For example, correcting the task-related BOLD signal using cerebrovascular reactivity via a hypercapnic challenge altered the BOLD signal in three ways: (1) uncorrected age-related decreases in the MTL and lingual gyrus were no longer associated with age after correction, (2) uncorrected associations with age now showed significant increases with age in some regions like the left prefrontal cortex, and (3) uncorrected age-related increases in the right prefrontal cortex now revealed highly significant positive associations with age after correction ([45]). Other research has used less-invasive techniques such as correcting the BOLD signal using resting-state fluctuation analyses (RSFA) that have been shown to be correlated with vascular-related physiological functioning rather than neuronal functioning. In one eloquent example, [78] ([78]) showed that RSFA was related cardiovascular function (e.g., heart rate and heart rate variability) but not neural activity estimated using magnetoencephalography. Furthermore, this study showed that RSFA scaling on task-related BOLD signal resulted in large reductions in age-related differences, suggesting that much of the previous research has overestimated the effects of aging on neural activity when using uncorrected BOLD signals. [46] ([46]) also showed that the BOLD signal was related to both cerebrovascular reactivity and RSFA (which were related to each other), providing independent and converging evidence that vascular functioning can modulate the BOLD signal. Together, these studies suggest that the BOLD signal is influenced by cerebrovascular (non-neural) signals in quite unpredictable ways.

When investigating the associations between non-neural physiological effects on task-related brain activity, the canonical HRF is often used. To the extent that cerebrovascular alterations are more likely among people with vascular health risks, task-related BOLD signals might be misinterpreted if uncorrected. It is currently unknown how individual differences in vascular risk profiles that often are exacerbated by age alter the response of the BOLD signal to task-evoked stimuli. Accounting for variations in the HRF due to age or vascular risk may reduce the potential for false positive or false negative results in task-related neural recruitment. If variations in the BOLD signal are not accounted for, inferences of the cognitive processes underlying fMRI results may not be valid, particularly in older persons with poorly characterized vascular health. For example, a common finding is an increase in brain activity with aging and in individuals with vascular risk, which has been interpreted as (attempted) compensation ([28]; [59]). However, such increases in brain activity may not be due to greater neural activity at all if they were caused by vascular changes. Thus, the interpretation of these findings could easily change from a “compensatory” account to either an account of “brain maintenance” if no differences in neural activity between groups exist or even an account of “dysfunction” if neural activity decreases with age.

We argue that it is critical to understand and limit the potentially confounding nature of vascular changes in fMRI studies. Rather than simply excluding subjects with possible cerebrovascular risks, thereby dramatically reducing the generalizability of any results to the larger population of older adults, a key question is whether researchers should calibrate individual subjects’ HRF based on their individual patterns of vascular risk or if utilizing a canonical HRF affords a better approach. That is, in light of the multivariate, interdependent nature of vascular risk factors, the best approach for accounting for elevated vascular risk on BOLD/HRF is unclear. This question is further complicated by evidence of inherent variability in the HRF among young adults, suggesting the problematic nature of a canonical HRF as a standard in fMRI, independent of age ([1]). Given the variability in the HRF in young, healthy populations, it is unsurprising that further variability would be seen in aging populations in which there is more likely to be alterations as well as pathologies in the vascular system. Overall, these findings suggest that studies investigating inter-individual or group differences might be particularly vulnerable to misinterpretations.

### 1.3. The Present Study

Here, we sought to model the aggregate effects of age-prevalent vascular risk influences on the shape and timing of BOLD signal. We also tested the extent that corrections in the HRF shape and timing would alter the results of investigations on aging and vascular risk on the estimations of brain activity using fMRI. We approached this issue by considering that an accumulation of vascular health risks might lead to a crucial tipping point between neuronal viability or its death ([20]; [52]). Thus, an increased number of risk factors likely have greater impact on the BOLD signal, potentially representing a cumulative sum of potential neuronal injury on the brain that represents multiple pathways ([31]; [79]). Thus, we recruited middle-aged and older adult subjects ranging in age from 50 to 75 with a range of risk factors previously associated with poor health and late-life cognitive decline and a young adult reference group.

We used a slow event-related design to estimate the shape and timing of each subjects’ HRF while subjects viewed a flashing checkerboard. We predicted that older age and greater vascular risk would result in a more delayed HRF and smaller HRF peaks. We then compared the effects of aging and cumulative vascular risk on the estimated brain activity during an associative memory encoding task when applying assumptions of the canonical HRF versus a subject-specific HRF (sHRF). In these analyses, we focused on the ability for subjects to activate MTL regions and deactivate regions within the default mode network (DMN) during the memory encoding task. DMN regions were chosen because of the role they play in memory ([37]; [68]; [70]), the age-related inability to deactivate such regions ([48]; [58]; [61]; [62]), and the potential link between DMN regions and the vascular system ([27]; [33]). In addition, both sets of regions have been shown to be altered by aging and vascular risk in studies using neuroimaging modalities other than fMRI ([5], [4]; [7]). We hypothesized that using an sHRF would result in greater sensitivity in detecting effects of age and vascular risks on the estimated brain activity than when using a canonical HRF. The reliability of each approach was assessed by separating testing two independent runs in a subset of participants. Lastly, using a whole-brain approach, we hypothesized that previously found increases in brain activity that often are interpreted as compensation or attempted compensation would be reduced after implementing an sHRF in each subject.

## 2. Materials and Methods

### 2.1. Participants

Data were collected as a part of the Alabama Brain Study on Risk for Dementia to investigate the influence of cumulative risk factors on brain structure and brain function in adults free of dementia. This study included the first 66 subjects from this research program: 20 younger adults aged 20–30 years (M = 23.30; 11 females), 25 middle-aged adults aged 50–62 (M_age_ = 55.16; 19 females), and 21 older adults aged 63–74 (Mage = 67.23; 14 females). Data from 1 subject in each age group (N = 3) could not be used because of excessive head motion in the scanner, resulting in a total sample of 63. In 9 additional subjects (M_age_ = 62.19; 6 females), the sHRF was calculated independently in two sessions to assess the reliability of the sHRF estimation. All subjects were recruited from the Tuscaloosa and Birmingham areas within Alabama. Younger adults were recruited through word of mouth whereas middle-aged and older adults were also recruited through flyers, Facebook ads, and newsletters. Middle-aged and older adults had a moderate to high level of cognitive functioning as measured by the St. Louis University Mental Status (SLUMS; M = 26.88, Range = 21 to 30; [75]).

In contrast to most middle-aged and older adult samples used in cognitive aging research that are often very healthy, middle-aged and older adults were recruited to vary health-related issues found in a more normative aging population and to enrich the sample with risks for dementia, which include vascular health risks (e.g., [20]; [21]; [38]). Inclusion criteria included being right-handed, having a SLUMS score above 19 (after adjusting for education), the ability to understand and comprehend English, and at least one of the following self-reported risks for dementia that were assessed over the initial phone screening: subjective memory complaints, less than a high school education, African American or Hispanic ethnicity, mild head trauma, family history of Alzheimer’s disease, current diagnosis of hypertension or systolic blood pressure greater than 140 mmHg, current diagnosis or a family history of heart disease, current diagnosis of high total cholesterol, history or current use of smoking tobacco, current diagnosis or family history of diabetes, and body mass index greater than 30 kg/m^2^. Exclusion criteria included being pregnant, having a prior diagnosis of dementia or another neurological condition, have a medical history of stroke or traumatic brain injury, claustrophobia, history of substance abuse, or have metallic implants that are incompatible with MRI. Vision was normal or corrected to normal using MR-compatible glasses or contact lenses. All subjects gave informed consent and were monetarily reimbursed for their time and participation in this study. The study was conducted in accordance with the Declaration of Helsinki, and the protocol was approved by the Institutional Review Board of the University of Alabama (#16-011-ME) on 20 July 2016.

### 2.2. Procedures

#### 2.2.1. Overall Study

The study was distributed across two in-person sessions. The first session consisted of a cognitive battery including the SLUMS and an assessment of health metrics. At the end of this session, subjects were provided with an online survey to fill out several questionnaires including health history. The second session consisted of the MRI scan that included structural scans and several fMRI scans.

#### 2.2.2. Health Questionnaire

In an online survey, subjects were asked several questions that were used in the Jackson Heart Study ([76]) regarding their personal health and family health history. Specifically, questions from the Personal and Family History Form were used to assess self-reported diagnosis of diabetes, family history of diabetes, hypertension, or dyslipidemia. For example, subjects were asked, “Has your doctor or health professional ever said that you have high blood pressure?” and answered identical questions to measure if they had prior diagnosis of either elevated blood cholesterol or diabetes. The Physical Activity Form was used to measure the frequency that subjects engage in physical activity during their leisure time; each item used a five-point scale: less than once a month, once a month, 2–3 times a month, once a week, and more than once a week.

#### 2.2.3. Health Metrics

An Omron HBF-516 scale was used to gather biometric information including body mass index, body fat, visceral fat, and resting metabolism. A HealthSmart semi-automatic blood pressure monitor was used to estimate systolic and diastolic blood pressure after being seated for at least 20 min from which we calculated pulse pressure (i.e., SBP-DBP). Abdominal circumference was assessed by using a measuring tape around one’s abdomen at the navel. Lastly, gait speed was measured by having subjects walk at a normal pace over a 10 ft distance while being timed ([71]; [81]). This task was conducted twice and times were averaged.

#### 2.2.4. fMRI Checkerboard Task

A reversing checkerboard (changing 8 times over 1 s) was presented for 1s in each of 20 repetitions. When the checkerboard was presented, the subject was instructed to view the image and tap the response button. Inter-trial intervals varied between 8 s and 16 s with an average of 12 s, during which the subject viewed a fixation cross. A total of 73 volumes were collected over a two-minute span. The purpose of this task was to obtain a measure of each sHRF in the occipital cortex that was time-locked to the onset of the visual stimulation. Nine participants had two sessions to assess reliability of the sHRF.

#### 2.2.5. fMRI Memory Task

In the scanner, subjects were asked to view a picture of a neutral face above a picture of either an object or scene for 3 s either on the left or right side of the screen. To assist their memory, subjects were told to imagine the person interacting with the object or within the scene. After viewing the pair of pictures, subjects were asked how likely they think they would remember the face-object or face-scene pair and rate their prediction on a three-point scale: likely to remember, may possibly remember, or unlikely to remember. Trials were separated by a fixation cross of jittered duration (1.72–17.20 s). Stimuli were divided into two runs with each run consisting of 32 pairs of pictures and lasting for 8 min, totaling 558 volumes.

During the retrieval task, subjects were presented with a previously seen face and asked to choose the correct object or scene pairing from five options. The options included pictures of two objects, two scenes, and a “never seen” option that could be chosen if subjects believed that they never saw the face. Of the four picture options, two were always in the object category and two were always in the scene category, but only one was previously paired with the face during the encoding session. The location of the pair (left or right side of the screen) was also provided to subjects for half of the trials to help cue their memory. The test was divided into two runs with each run consisting of 32 memory trials and lasting for 5 min.

The face stimuli were taken from the Chicago Face Database Version 2.0.3 that includes high-quality photos of male and female faces from different ethnoracial categories ranging in age from 18 to 50 years old ([49]). Half of the faces chosen were of men, half were of women and within each sex, half were African American and half were non-Hispanic White. The ethnoracial categories were chosen to represent the local demographics. The pictures of objects and scenes were taken from http://olivalab.mit.edu/MM/stimuli.html (accessed on 4 July 2025) and have been used in several studies to assess memory ([39]; [40]; [67]).

The order of items and fixations was maximized for event-related fMRI using optseq2 program (https://surfer.nmr.mgh.harvard.edu/fswiki/optseq2; accessed on 7 March 2017). One order was created for each memory task run, totaling 4 runs (2 encoding and 2 retrieval). These four orders were rotated across retrieval cue type (left, right, no cue) across subjects.

### 2.3. Data Analyses

#### 2.3.1. Vascular Risk Score Calculation

Our vascular risk factors were submitted to an exploratory factor analysis using direct oblimin rotation. The risk factors included the presence of diabetes, high total cholesterol, hypertension, obesity and estimations of arterial stiffness, visceral fat, abdominal circumference, body fat, resting state metabolism, engagement in physical activity, gait speed, and body mass index. The number of factors was determined through a Parallel Analysis ([80]) in which a matrix of random and uncorrelated data was compared with a matrix of the original data. The eigenvalues were then computed for each of the two correlation matrices. Factors were retained when the eigenvalue of the factor from the original data matrix was greater than the eigenvalue of the factor from the randomly generated data. If the eigenvalues from the randomly generated data were equal to or greater than the eigenvalues from the original data matrix, it was assumed to be due to noise. The factor loadings from this analysis were used to form a weighted average on each standardized factor to form separate vascular risk factors.

#### 2.3.2. fMRI Acquisition and Preprocessing

A 3T Siemens PRISMA scanner was used to collect MRI data at the UAB Civitan International Neuroimaging Laboratory. Structural scans were acquired using high resolution T1-weighted structural MP2RAGE (parallel acquisition acceleration type = GRAPPA; acceleration factor = 3, TR = 5000 ms, TE = 2.93 ms, TI 1 = 700 ms, TI 2 = 2030 ms, flip angle 1 = 4 degrees, flip angle 2 = 5 degrees FOV = 256 mm, matrix = 240 × 256 mm^2^, in-plane resolution = 1.0 × 1.0 mm^2^). For functional scans, T2*-weighted images were used to estimate neural activity via the BOLD signal (56 interleaved axial slices, 2.5 mm thickness) using an EPI sequence (TR = 1720 ms, TE = 35.8 ms, flip angle = 73 degrees, FOV = 260 mm, matrix = 104 × 104 mm, in-plane resolution = 2.5 × 2.5 mm^2^, multi-band acceleration factor = 4). Data were preprocessed using Statistical Parametric Mapping 12 (SPM12; https://www.fil.ion.ucl.ac.uk/spm/; accessed on 4 July 2025). Preprocessing steps included calculating and correcting for head movement, coregistering the structural images to the functional images, segmenting the structure into gray matter, white matter, and cerebral spinal fluid, normalizing to the MNI template (2 mm cubic voxels), and spatial smoothing using 6 mm full-width half-max (FWHM) kernel. Artifact Detection Tools was used to outliers due to movement or signal intensity spikes ([51]).

#### 2.3.3. Statistical Analyses

To estimate each subjects’ sHRF, a finite impulse response function was used to model task-evoked brain activity in response to the flashing checkerboard across the whole brain. In this general linear model (GLM), 12 basis functions were modeled across a 21 s window and outliers were entered as nuisance regressors. At the first level, an F-test was calculated across the 12 time points at a liberal threshold of *p* = 0.05 using an inclusive mask of the occipital cortex. At the peak voxel within this mask, parameter estimates were extracted, which were used to represent the sHRF. For each subject, the following metrics were calculated: maximum value of the sHRF, the latency (the time point at which that maximum value occurred), and the FWHM of the sHRF (the distance between the rise and fall of the HRF at the point at which the curve has reached half of its maximum). Note that the area under the curve of the HRF was also calculated, but was highly correlated with the maximum value of the HRF (r(61) = 0.86, *p* < 0.001) and so only the maximum value is reported for simplification.

In the 9 subjects that had two checkerboard runs, Cronbach’s Alpha was calculated for each of the 12 time points in the first run and compared to the 12 time points in the second run. Because the general shape and timing of the sHRF would be generally similar across all individuals, a cross-subject Cronbach’s Alpha was also calculated for each subject by using the 12 time points in one run and comparing it to the 12 time points in a different subject’s HRF. While all scores were expected to be high, a higher score for the within-subject Cronbach’s alpha than the cross-subject Cronbach’s alpha would suggest that the estimated sHRF would be truly unique to an individual. Regression analyses were also conducted using age, vascular risk, and their interaction on the HRF metrics after centering each variable separately for each vascular risk factor.

For the memory encoding task, we first conducted first level analyses under the assumptions of the GLM using a stick function and the canonical HRF with the addition of both temporal and dispersion derivatives. Two trial types of interest were included as regressors in this analysis: trials for which subjects subsequently remembered the picture pair and trials for which the subjects did not correctly remember the picture pair. Other regressors of non-interest include outliers and session effects. We next conducted a first level analysis substituting the canonical HRF (in the variable SPM.xBF.bf of the SPM.mat) with each subject’s sHRF (which was expanded to correspond to the appropriate microtime resolution). The Matlab script used to replace the canonical HRF with each subject’s HRF can be found at https://github.com/immcdonough/SPM_sHRF (accessed on 20 October 2025).

At the second (group) level, we first used a region of interest (ROI) approach to investigate differences in the estimations of brain activity when using the canonical HRF compared with the sHRF. ROIs were chosen based on previous (non-fMRI) studies investigating aging and vascular risk on the brain including bilateral MTL (hippocampus and parahippocampal gyrus) and DMN (posterior cingulate cortex (PCC), ventromedial prefrontal cortex (vmPFC) regions. Freesurfer was used to create the ROIs from the MNI template. For each ROI, parameter estimates were extracted for correct trials (versus baseline) and averaged across hemispheres. Mixed analyses of covariance were conducted by entering analysis type (canonical HRF, sHRF) as a within-subject variable, age, vascular risk, and their interactions as between subject variables. Brain activity in each of the ROIs was the outcome variable for each analysis. All variables were entered as continuous measures. Arbitrary age group divisions are for visualization purposes only.

The alpha level was set to 0.05. We also conducted several whole-brain analyses. A paired t-test based on the analysis type (canonical HRF vs. sHRF) on the correct trials (versus baseline) to understand the global effects of using the sHRF. Lastly, regression analyses were conducted to test age × vascular risk interactions on brain activity separately for each analysis type across the whole brain. An alpha was set to 0.001 with 100 contiguous voxels for each whole-brain analysis.

## 3. Results

### 3.1. Participant Demographics

Participant demographics can be found in Table 1. Chi-square tests and correlations were used to test age differences between demographic, vascular risk, and cognitive factors. Middle-aged and older adults were more likely to have a family history of diabetes than younger adults (χ^2^ = 12.688, *p* = 0.002). Older age was associated with a higher likelihood of having hypertension (χ^2^ = 16.233, *p* < 0.001). Older age was also associated with greater arterial stiffness (r = 0.492, *p* < 0.001), more visceral fat (r = 0.367, *p* = 0.003), greater abdominal circumference (r = 0.267, *p* = 0.0357), slower gait speed (r = 0.309, *p* = 0.015), and poorer memory performance on the fMRI task (r = −0.642, *p* < 0.001).

### 3.2. Factor Analysis of Vascular Risk Factors

The factor analysis yielded three factors that cumulatively explained 56.6% of the variance in the data. The first factor was labeled Metabolic Impairment and consisted of resting metabolism (1.040), visceral fat (0.643), and abdominal circumference (0.543). This first factor explained 19.80% of the covariance. The second factor was labeled Physical Inactivity and consisted of body fat (0.958), body mass index (0.616), obesity (0.557), physical activity (−0.520), and gait speed (0.383). This second factor explained 19.90% of the covariance. The last factor was labeled as Diabetic Syndrome and consisted of high total cholesterol (0.793), hypertension (0.654), diabetes (0.590), family history of diabetes (0.499), and arterial stiffness (0.399). This factor explained 16.90% of the covariance. Furthermore, each factor was correlated with one another ranging from r = 0.39 to r = 0.68 (all *p*’s < 0.002).

### 3.3. Age and Risk Effects on HRF Metrics

As seen in Figure 1, the HRF shape was quite variable across subjects. Within-subject Cronbach’s alpha scores ranged from 0.79 to 0.98 and had an average of 0.90, suggesting an excellent average internal consistency. Cross-subject Cronbach’s alpha scores ranged from 0.49 to 0.99 and had an average of 0.81, suggesting a good average internal consistency. *T*-tests comparing within-subject and cross-subject Cronbach’s alpha were significant (*p*’s < 0.01), suggesting that our sHRF estimation was reliable and provided unique information about the shape and timing of the sHRF for each individual.

We next investigated how age and vascular risk factors were associated with the shape and timing of the HRF assessed during the checkerboard task. These HRF metrics included maximum value, latency, and FWHM. Separate tests were conducted for each of the risk factor scores. For the maximum HRF value, we found a main effect of Age (b = −0.004, SE = 0.001, *p* < 0.001), a marginal main effect of Metabolic Impairment (Factor 1) (b = 0.170, SE = 0.098, *p* = 0.088), and no interaction (b = −0.003, SE = 0.006, *p* = 0.559). As can be seen in Figure 2, the main effect of Metabolic Impairment appeared to be driven by the middle-aged adults.

For the second factor, we found a main effect of Age (b = −0.004, SE = 0.001, *p* = 0.002), but no main effect of Physical Inactivity (Factor 2) (b = 0.135, SE = 0.100, *p* = 0.185), and no interaction (b = −0.003, SE = 0.007, *p* = 0.665). Although Figure 3 shows the possibility of an interaction between age and Physical Inactivity, the higher variability rendered this interaction quite non-significant. Therefore, older age was associated with a smaller peak maximum value of the HRF for the first two factors with little or no additional influence on vascular risk on the HRF.

However, for the Diabetic Syndrome Risk (Factor 3), we not only found a main effect of Age (b = −0.008, SE = 0.002, *p* < 0.001), we also found a main effect of Diabetic Syndrome Risk (b = 0.469, SE = 0.160, *p* = 0.005), and an Age × Diabetic Syndrome Risk interaction (b = −0.022, SE = 0.009, *p* = 0.024). As shown in Figure 4, the interaction was due to the fact that greater risk was associated with greater peaks of the HRF for younger adults (b = 0.999, SE = 0.345, *p* = 0.005) and middle-aged adults (b = 0.327, SE = 0.136, *p* = 0.019), but not older adults (b = 0.045, SE = 0.162, *p* = 0.782).

For latency and FWHM of the HRF, no effects were found for Age and Vascular Risk, nor their interactions across the three different factors (all *p*’s > 0.05).

### 3.4. Age, Risk, and Analysis Type Effects on the Estimates of Brain Activity in the ROIs

We next assessed how using the sHRF impacted the estimation of the GLM and subsequent relationships with Age and Vascular Risk on brain activity. Not all brain regions were equally impacted by the different models of the HRF. When modeling correct trials only, some regions showed nearly identical time series after deconvolved with the HRF (Figure 5A), whereas other regions showed a greater amplitude when using the sHRF (Figure 5C). Additionally, when assessing the peri-stimulus time histogram for each of these regions, we can see how the sHRF can model mean activity differently from the canonical HRF (Figure 5B,D).

A priori ROIs in MTL (hippocampus, parahippocampal gyrus) and DMN (vmPFC, posterior cingulate cortex) show further differences between the two HRFs. Figure 6 illustrates the relationship between the parameter estimates in each of the analyses using a scatterplot for each of the ROIs with the red line representing an identity line, or the case for a perfect 1 to 1 correspondence. As can be seen across the four ROIs, using the sHRF did not change the parameter estimates for some of the subjects (i.e., their data point fell on the identity line). Other individuals had data points far away from the identity line, indicating a large change in the parameter estimates as a result of the change in analysis. When a change did occur, it was more likely to occur in the direction of canonical HRF > sHRF for the MTL regions and sHRF > canonical HRF for the DMN regions (especially, the vmPFC). We quantified these observations with regression analyses to better understand which subjects were more likely to exhibit these changes.

In the hippocampus, we found only a main effect of Analysis Type for each of the three risk factors (Factor 1: F(1,59) = 18.188, MSE = 0.070, *p* < 0.001, η_p_^2^ = 0.236; Factor 2: F(1,59) = 15.258, MSE = 0.070, *p* < 0.001, η_p_^2^ = 0.205; Factor 3: F(1,59) = 7.473, MSE = 0.070, *p* = 0.008, η_p_^2^ = 0.112). In all cases, using the sHRF resulted in lower levels of brain activity than when using the canonical HRF (Figure 7).

In the parahippocampal gyrus, we similarly found main effects of Analysis Type for each of the three risk factors (Factor 1: F(1,59) = 8.576, MSE = 0.090, *p* = 0.005, η_p_^2^ = 0.127; Factor 2: F(1,59) = 8.576, MSE = 0.091, *p* = 0.005, η_p_^2^ = 0.127; Factor 3: F(1,59) = 6.197, MSE = 0.082, *p* = 0.016, η_p_^2^ = 0.095). However, for Factor 3 (Diabetic Syndrome Risk) we also found a main effect of Age (F(1,59) = 5.503, MSE = 3.389, *p* = 0.022, η_p_^2^ = 0.085), and Analysis Type × Risk interaction (F(1,59) = 5.644, MSE = 0.082, *p* = 0.021, η_p_^2^ = 0.087). This interaction was due to a stronger positive relationship between Diabetic Syndrome Risk and brain activity in the canonical HRF analysis (b = 3.059, SE = 1.433, *p* = 0.037) than in the sHRF analysis (b = 2.030, SE = 1.378, *p* = 0.146).

In the vmPFC, we found main effects of Analysis Type for each of the three risk factors (Factor 1: F(1,59) = 9.393, MSE = 0.123, *p* = 0.003, η_p_^2^ = 0.137; Factor 2: F(1,59) = 9.952, MSE = 0.123, *p* = 0.003, η_p_^2^ = 0.144; Factor 3: F(1,59) = 22.368, MSE = 0.105, *p* < 0.001, η_p_^2^ = 0.275). Unlike the previous ROIs, these main effects were due to greater levels of brain activity when using the sHRF compared with the canonical HRF (see Figure 7). Note that because this region showed deactivations during memory encoding, this finding can be interpreted as less deactivation after using the sHRF. In addition to these main effects, for Factor 3 (Diabetic Syndrome Risk) we found an Age × Risk × Analysis interaction (F(1,59) = 11.887, MSE = 0.105, *p* = 0.001, η_p_^2^ = 0.168). As shown in Figure 8, this was due to a larger Age × Risk interaction using the canonical HRF than in the sHRF. Specifically, as vascular risk increased greater deactivations in the vmPFC were found for younger adults, a smaller relationship was found for middle-aged adults, and the relationship reversed older adults. This interaction was minimized after using the sHRF.

In the posterior cingulate cortex, no main effects or interactions were found.

### 3.5. Comparison of Canonical HRF vs. Subject-Specific HRF During Encoding

To assess the global effects of using an sHRF, we conducted a whole-brain analysis comparing successful encoding activity > baseline. Consistent with the ROI analysis, we found that task-positive regions exhibited a reduction in brain activity when using the sHRF compared with the canonical HRF (see Figure 9). We found that task-negative regions exhibited increased brain activity when using the sHRF compared with the canonical HRF (i.e., less deactivation).

### 3.6. Age and Risk Effects Using Canonical vs. Subject-Specific HRF: Whole Brain Analysis

Appendix A report the whole-brain regression analyses when using the canonical HRF and the sHRF, respectively. We will first report the age effects, followed by the vascular risk effects for each of the three factors, and then the interactions. For each analysis, the primary focus will be on the qualitative differences between the canonical and the sHRF results.

For the effects of age, positive associations between age and brain activity were found for bilateral supramarginal gyrus using the canonical HRF when both Factor 1 and Factor 2 were in the regression models with the T-statistics ranging from 4.31 to 5.82 and cluster sizes ranging from 107 to 334. The same regions were found when using the sHRF with the T-statistics ranging from 4.75 to 5.23 and cluster sizes ranging from 149 to 327. No notable differences were found in this contrast between the two analyses. Negative associations with age were also found when Factor 1 was included in the model in left caudate and left paracentral lobule when using the canonical HRF. When using the sHRF, the left paracentral lobule was also significant with a greater statistical magnitude (5.58 vs. 4.56) and cluster size (623 vs. 142) compared with the canonical HRF analysis. However, the left caudate was no longer significant when using the sHRF, and instead an area of the left cingulate cortex was now significant, thus resulting in qualitatively different significant regions depending on the analysis type. Interestingly, while the left paracentral lobule was not significant at the given threshold when using the canonical HRF and Factor 2 or Factor 3 were in the model, this region was significant when Factor 2 or Factor 3 were in the model when using the sHRF, suggesting that using the sHRF resulted in more consistent significant results at the current threshold regardless of the risk factors entered into the regression model.

Neither positive nor negative effects of vascular risk factor were significant for any of the analyses. However, interactions between Age × Vascular risk were present. Age × Vascular Risk interactions with positive beta weights were only found for Factor 3 when using the canonical HRF and occurred in the right anterior cingulate cortex. The analogous regression using the sHRF had no significant regions. We also found an Age × Vascular risk interaction with a negative beta weight for Factor 1 in the left middle temporal gyrus when using the canonical HRF that was also present when using the sHRF. The magnitude and cluster size was larger when using the sHRF than the canonical HRF for this region (6.65 and 184 vs. 6.05 and 107). When using the sHRF and Factor 1, two additional regions were significant including the left caudate and the left cerebellum. As can be seen in Figure 8, these interactions revealed a positive association between Factor 1 risk and brain activity for younger adults, but a negative association in middle-aged and older adults. The only other Age √ Risk interaction occurred for Factor 2 when using the sHRF in the left cuneus. This interaction took the same form as the other previous interactions with Factor 1.

### 3.7. Brain–Behavior Correlations

To determine whether some of the conflicting analyses could be due to a reduction in variance in the BOLD signal (adding meaningful clusters or removing non-meaningful clusters), we correlated those clusters with the in-scanner memory performance. To the extent that brain activity in this memory task was behaviorally relevant, one would predict that a “true” signal would be more likely to be associated with memory performance and that a “false” signal would not. At a minimum, assessing such brain–behavior correlations offers a converging test as to how using the canonical HRF versus the sHRF may affect individual difference analyses.

When using our ROIs, we found that memory performance was correlated with brain activity in the hippocampus in both analyses (r = 0.334, *p* = 0.007 and r = 0.321, *p* = 0.010 for the canonical HRF and sHRF, respectively), was only significantly correlated with parahippocampal gyrus brain activity in the sHRF analysis (r = 0.232, *p* = 0.067 and r = 0.273, *p* = 0.030 for the canonical HRF and sHRF, respectively), and was not correlated with the posterior cingulate cortex or vmPFC in either of the analyses (all *p*’s > 0.10).

We next correlated memory performance with clusters that differed between the analyses in the whole-brain analyses (see Appendix A). Two clusters within the anterior cingulate cortex were associated with Age × Vascular Risk (Factor 3) interactions when using the canonical HRF, but not the sHRF. No correlation was found between memory performance and brain activity in either of these two clusters (all *p*’s > 0.29). Three other clusters exhibited interactions when using the sHRF, but not when using the canonical HRF, which included the caudate, the cerebellum, and the cuneus. None of these clusters were significantly associated with memory performance (all *p*’s > 0.09). Lastly, a cluster in the left paracentral lobule was negatively associated with age in one of the three Age × Vascular Risk analyses when using the canonical HRF but was consistently found in all three of the Age × Vascular Risk analyses when using the sHRF. This cluster was significantly associated with memory performance (r = 0.325, *p* = 0.009).

## 4. Discussion

The present study aimed to investigate how aging and vascular risk affected the HRF to task-evoked stimuli, and whether the use of a subject-specific HRF would be preferable to the use of the standard canonical HRF. Towards these aims, we analyzed whether the relationships between age, vascular risk, and memory performance would differ when taking into account subject-specific differences in HRF shape and timing. We found that the shape of the HRF differed more with age than it did with vascular risk. Specifically, we found that older age was associated with a smaller maximum HRF peak. No age-related differences were found in other HRF metrics such as time to peak (latency) or the width of the HRF. Greater diabetes-related risk factors were associated with a higher HRF maximum peak in younger and middle-aged individuals, but not older adults. In task-related areas, the use of the subject-specific HRF yielded lower average levels of brain activity compared to the use of the standard canonical HRF. Nevertheless, these differences were not true in all subjects. We found that the correlations between brain activity and memory performance on the fMRI task were largely unchanged when analyzing the data using the canonical versus the subject-specific HRF. Overall, such subject-specific adjustments should presumably reduce variance and increase accuracy (e.g., [54]), but our findings alone do not yet provide sufficient evidence for the latter.

Our finding that older age was associated with smaller maximum HRF peaks differs from an early study by [32] ([32]). Huettel et al. used a checkerboard task similar to the one that we used to assess the HRF in the visual cortex but found no differences in the peak of the HRF in older adults. Instead, they found that the latency of the peak was faster in older adults than younger adults. Several methodological differences might explain the discrepancy in findings. Our study recruited a larger number of subjects (63 vs. 22), used a 3T scanner and thus better signal-to-noise ratio (vs. a 1.5T scanner), and our repetition time was slower (TRs: 1.72 s vs. 1.0 s). Despite differences between our findings and Huettel et al., the common theme is that there can be subject-specific individual differences in the HRF.

Interestingly, we found that a collection of risks related to diabetes was associated with a larger maximum peak of the HRF in young and middle-aged adults and not older adults. This finding suggests that vascular risks impact the shape of the HRF in presumably normal adults, thereby putting doubts on the claim that one does not have to worry about differences in the HRF in young adults. The direction of the effects also deserves some scrutiny; our findings were in the form of apparent increases in brain activity in those with greater vascular risk. Nevertheless, the notion of a decoupling between the BOLD signal and neural activity has been attributed to apparent decreases in brain activity in past research (e.g., [19]). It has been argued that increases in brain activity, as in our findings, are arguably more difficult to explain by such a decoupling (cf. [64]). On the other hand, [28] ([28]) suggested that much of the apparent increased activity found in older adults or adults at risk for Alzheimer’s disease might incorrectly be attributed to a compensatory process when this increased activity might be due to the confounding effect of vascular differences on the BOLD signal. In other words, their apparent increased activity may in fact be partially an error of measurement. In line with this concern, [59] ([59]) found that increased brain activity in the frontal cortex with increasing age was likely due to a reduction in CMRO2 rather than an increase in neural activity. They concluded that at least in the context of their study, the increase in brain activity was related to a reduction in neural activity. Thus, if uncorrected, the increases in the peak of the HRF that was associated vascular differences in our study could be (incorrectly) interpreted as compensatory increases in brain activity in individuals with high vascular risk. Because research has supported the notion that both apparent increases and decreases in brain activity could be due to non-neural vascular differences between groups, one cannot use direction of the brain effects alone to infer compensatory or declining brain functioning. We argue that correcting potential cerebrovascular changes yields more accurate information of aging and health but not necessarily always in a single direction. Indeed, other studies using this technique also have found both stronger and weaker associations that are brain-region dependent (e.g., [54]).

To test whether variability in HRFs affect task-related fMRI, we estimated brain activity when using the canonical HRF in a commonly used neuroimaging software package (SPM) compared to each subject’s own HRF (sHRF). Notably, we also included temporal and dispersion derivatives in the canonical HRF model, which many researchers include to account for such variability in the HRF. Using the sHRF resulted in a “dampening” of apparent neural activity such that task-positive regions exhibited activity closer to zero and task-negative regions exhibited activity closer to zero (compared to our canonical HRF analysis). This evidence, again, is consistent with the notion that brain activity would normally be interpreted as higher if the standard canonical HRF model were to be used. We speculate that this apparently higher level of brain activity could be partially confounded by not using a precise enough HRF within each subject.

Critically, not all subjects exhibited such an apparent dampening of brain activity, suggesting that using the sHRF did not lead to a unidirectional systemic bias in the analysis. Rather, some of these changes could have been due to outliers—extreme values of the parameter estimates, that were then decreased. For example, one subject’s parameter estimate in the hippocampus decreased in size from 4.33 using the canonical HRF to 3.49 using the sHRF (average parameter estimate was 0.58), a difference of 0.84, whereas the average change in the parameter estimates in the hippocampus was only 0.21. Similarly, in the vmPFC one subject’s parameter estimate changed in size from −6.23 using the canonical HRF to −4.77 using the sHRF (average parameter estimate was −0.44), a difference of 1.46 when the average change in this region was 0.19. Thus, one interpretation of these findings is that using the sHRF reduced extreme estimates of brain activity and that some of these extreme estimates were due to an HRF that deviated from the canonical HRF. This reduction in outlier values could be one reason why future researchers might consider using sHRF. The differences across brain regions also underscores the point that using an sHRF does not universally impact each brain region. We also show that the predicted timeseries in some brain regions are identical to the canonical HRF, also consistent with other measures correcting the BOLD signal (e.g., [45]; [78]).

To the extent that such evidence of “dampening” extreme values can be interpreted as minimizing variance due to error, one might then predict that using the sHRF would result in an increased ability to detect “true” effects. When estimating the effects of age and vascular risk on brain activity during memory encoding, we found evidence consistent with this claim. Using the sHRF resulted in more consistent and larger effects than when using the canonical HRF. Specifically, the magnitudes of the effects were often larger and the cluster sizes were also larger when using the sHRF compared with the canonical HRF. In the cases where the regions differed between the two analysis types, one analysis found two clusters in which significance was reduced below the current alpha threshold when using the sHRF and four analyses found one or two clusters in which significance was increased above the alpha threshold. Without overinterpreting which result corresponds with a “truer” effect, this finding at least signals a qualitative shift in the inferences one would make with the purported impacts of age and vascular risks on brain activity. One potential solution is for future research to include both sHRF and canonical HRF analyses until a consensus is found or a comparison with a gold standard is used to know which is most accurate.

Lastly, we found that the correlations between brain activity and memory performance were largely unchanged when using the canonical versus the subject-specific HRF. That is, the effect sizes of all of the activity-behavior correlations were quite similar. Upon inspection of Figure 6, this result should not be too surprising because the rank order of the parameter estimates across individuals largely remained intact for each ROI. However, the subtle shifts in individual differences did lead to some different results, which might lead one to argue in favor of using the sHRF over the canonical HRF. For example, activity in the parahippocampal gyrus was correlated with similar positive effect sizes with memory performance for both analyses, but the effect increased in size and became statistically significant when the sHRF was used and happened to pass the arbitrary *p* = 0.05 boundary. Additionally, the age effect in the left paracentral lobule was consistently found only when using the sHRF and this region showed a significant association with memory performance. One might take these latter findings as evidence that using the sHRF to estimate brain activity helped moderately increased the behavioral validity of the findings. Nevertheless, the differences between canonical and subject-specific HRF on these memory measures were smaller in other comparisons, so more empirical evidence is needed before one can firmly conclude that using sHRF is a more valid technique.

### 4.1. Alternative Interpretations

In the present study, we have no data that speaks to the “ground truth” of neural activity to verify that using the sHRF leads to better accuracy in measuring neural activity. Therefore, it is unclear whether better inferences about aging and vascular risk come when using subject-specific HRF in neuroimaging techniques. An alternative interpretation is that the sHRF leads to a poorer model fit of how neural activity is manifested in the BOLD signal, thus resulting in smaller parameter estimates (on average) almost entirely across the whole brain. This possibility could occur because we estimated each subject’s HRF in one voxel within the occipital cortex and, if the shape or timing of the HRF differs in other parts of the brain from the occipital cortex (which is quite likely), then the rest of the brain would result in poorer model fits. In contrast, the temporal and dispersion derivatives when the canonical HRF is used might provide sufficient flexibility to account for regional differences in the HRF. An ideal solution would be to model the subject-specific HRF in every voxel (or region) of the brain. However, to do this, one would need a separate task for every voxel of the brain that is maximally sensitive to that voxel (or region). This method is not practical nor is it known to which task each voxel or region of the brain responds maximally. Moreover, previous research has found that inter-individual differences in the HRF are much greater than intra-individual differences ([29]; [56]). This evidence suggests that estimating the HRF in one voxel for each subject and assuming a similar shape/timing in other voxels should explain more of the variability in the data than estimating the HRF in every voxel of the brain for one person and assuming the HRF is the same in all other individuals. Our argument is that because individuals differ in cerebrovascular health, these inter-individual differences in general HRF shape or timing will lead to the most accurate estimations of task-evoked brain activity, regardless of the brain region.

### 4.2. Implications for Studies on Aging, Health, and Disease

The process of aging is a complex phenomenon that is impacted by a host of factors that put cerebrovascular health at risk. These risks include hypertension, diabetes, dyslipidemia, obesity, and smoking (among many others). Because these risks have repeatedly shown to be associated with poorer cognition ([25]; [43]; [60]) and dementia ([34]; [66]; [72]; [85]), researchers have an increased interest in understanding how these factors impact brain function. One common suggestion is that those in various risk groups have an inability to deactivate brain regions within the DMN during memory encoding tasks (e.g., older adults: [48]; [58]; [61]; [62]; greater vascular risk: [10]; [16]; [50]; [84]; greater Alzheimer’s disease pathology: [69]; [82]; patients with Alzheimer’s disease: [13]; [48]; [63]). These reduced deactivations—i.e., higher than normal activation—have been considered a tell-tale sign of early neural dysfunction. Nevertheless, because subjects in these risk groups often have vascular health impairments, these findings may be in part due to differences in the blood vessels or blood flow in the brain (which provides a confound via HRF shape) rather than true differences in neural activity. In this study, we provided evidence that the vmPFC (a region within the DMN) ostensibly exhibited stronger deactivations with increasing vascular risk in older adults when using the canonical HRF. However, when the sHRF was used, these deactivations were also much weaker and no longer significant. Consistent with this general notion, [6] ([6]) showed that increases in vascular risk among older adults were associated with accelerated longitudinal decline in cerebral blood flow in DMN regions, further shedding doubt on increases in DMN regions in people with greater risk. We note that our sample size was small and thus we are showing a proof-of-concept rather than trying to generalize claims across all people with vascular risk factors.

While much of the research investigating fMRI activity as a function of age, health, and disease has occurred within cross-sectional samples, the present results also influence the interpretation of longitudinal or intervention studies using fMRI. For example, exercise or diet interventions may improve cerebrovascular functioning, that in turn results in changes in the BOLD signal (e.g., [9]; [18]; [44]). However, those BOLD signal changes might be misinterpreted as neuroplasticity, when in fact they are due to vascular changes in the brain which affect the HRF shape or timing ([83]). A possible solution to this issue is to utilize a subject-specific HRF.

### 4.3. Alternative Methods to Calibrate the BOLD Signal

Although we implemented a correction to the shape and timing of the HRF, we did not incorporate other methods to calibrate the BOLD signal shown also to be effective. To control for such vascular confounds, some studies have recommended a breath-hold/hypercapnia challenge to calibrate the BOLD response ([3]; [46], [45]; [65]; [77]) or resting state fluctuation amplitudes (RSFA) to calibrate the estimates of brain activity ([35]; [36]; [78]). We did not implement any of these methods in the current study. However, these solutions have not tested the extent to which the HRF shape or timing is impacted after these techniques are implemented. For instance, [78] ([78]) used the canonical HRF and implemented the RSFA adjustment after the first level models had been conducted, thus leaving open the possibility that differences in HRF shape still impacted the ability of the models to capture brain activity. Converging with this idea, it may not be sufficient to simply control for age or vascular risk factors as a covariate in the analyses due to non-linear influences that such factors have on the BOLD signal. More research is needed to test which correction methods (i.e., using an sHRF, resting state fluctuation amplitudes, etc.) or combination of correction methods would best correct for cerebrovascular contributions to the BOLD signal.

## 5. Conclusions

The present study provided evidence that both aging and vascular risk can be associated with differences in the shape of task-evoked fMRI activity. We also showed that this inter-individual variability in the HRF shape can alter statistical estimates of brain activity for both task-positive regions and task-negative regions during a memory task, leading to both decreases in group differences in some regions and increases in group differences in other regions. These differences can influence the conclusions that researchers and clinicians come to when using fMRI to examine the mechanisms underlying and treatments for health and disease across the adult lifespan. Thus, at least some of the interpretations made in previous studies using fMRI may be due to factors that influence the HRF that are non-neural. One solution that has been frequently used is to explicitly exclude subjects with high levels of vascular risk (e.g., diabetes, obesity, hypertension). However, this method leads to investigating so-called “super agers” rather than the original intended purpose of studying normal aging because normative aging is characterized by an increase in vascular risk. Indeed, some researchers have already started to recruit more heterogeneous samples in fMRI studies to make inferences on brain activity more generalizable to the population. They have performed this by relaxing recruitment exclusion criteria or using recruitment strategies that appeal to a broader range of individuals from different socioeconomic or ethnoracial backgrounds (e.g., [15]; [57]). While such generalization is critical, it should be performed so with caution to correct for the shape and timing of HRF when inter-individual correlational or group comparisons are being made. Using fMRI in combination with other neuroimaging techniques that do not rely on blood flow (e.g., magnetoencephalography; see [78]) might provide a more balanced approach not only to understanding how age and vascular risk impact the BOLD signal, but also to investigate the relationship between aging, vascular disease, and neural dysfunction (e.g., [54]).

## Figures and Tables

**Figure 1 behavsci-15-01457-f001:**
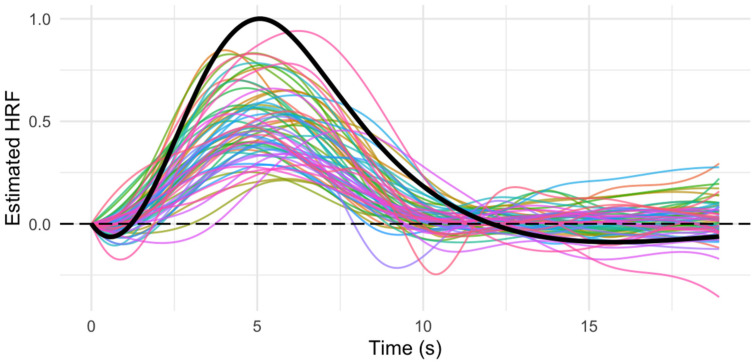
Plot of the hemodynamic response function (HRF) estimated for each subject using a finite impulse response analysis during a checkerboard visual stimulation task (multicolored lines) compared with the canonical HRF in SPM12 (black line).

**Figure 2 behavsci-15-01457-f002:**
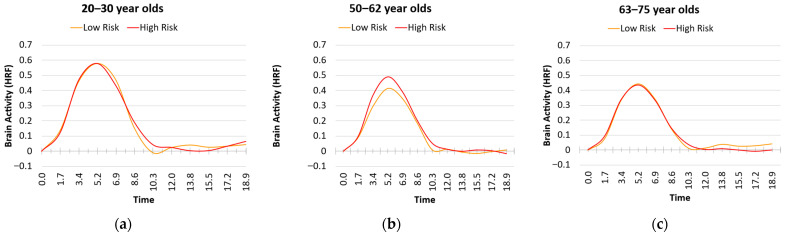
Plots of the hemodynamic response function (HRF) as a function of age group and vascular risks associated with Factor 1, Metabolic Impairment (Median Split: Low = Orange, High = Red): (**a**) Young adults with high risk show no difference in HRF peak than those with low risk; (**b**) Middle-aged adults with high risk show a higher HRF peak than those with low risk; (**c**) Older adults with high risk show no difference in HRF peak than those with low risk.

**Figure 3 behavsci-15-01457-f003:**
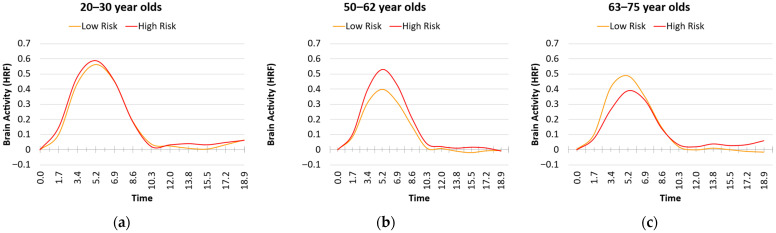
Plots of the hemodynamic response function (HRF) as a function of age group and vascular risks associated with Factor 2, Physical Inactivity (Median Split: Low = Orange, High = Red): (**a**) young adults with high risk show a slightly higher HRF peak than those with low risk; (**b**) middle-aged adults with high risk show a higher HRF peak than those with low risk; (**c**) older adults with high risk show a lower HRF peak than those with low risk.

**Figure 4 behavsci-15-01457-f004:**
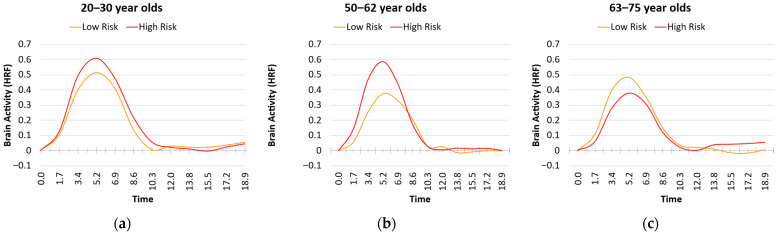
Plots of the hemodynamic response function (HRF) as a function of age group and vascular risks associated with Factor 3, Diabetic Syndrome (Median Split: Low = Orange, High = Red): (**a**) young adults with high risk show a higher HRF peak than those with low risk; (**b**) middle-aged adults with high risk show a higher HRF peak than those with low risk; (**c**) older adults with high risk show a lower HRF peak than those with low risk.

**Figure 5 behavsci-15-01457-f005:**
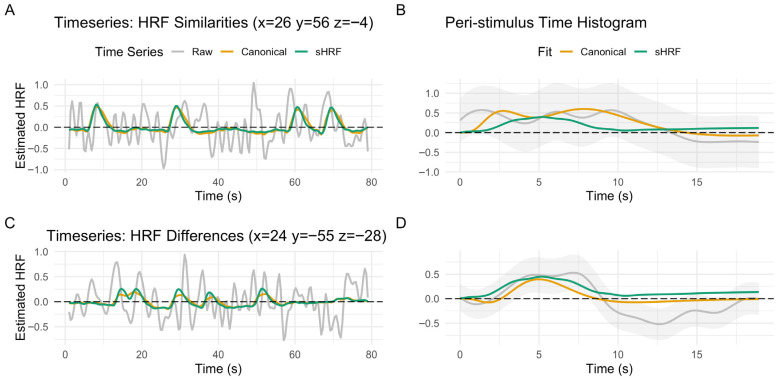
Plots of the hemodynamic response function (HRF) for correct trials over time (Raw Timeseries = Gray, Canonical HRF = Orange, sHRF = Green): (**A**) in a selected region showing similarities between the two HRF models; (**B**) peri-stimulus Time histogram of the same voxel in panel A as a function of HRF type (95% confidence intervals shaded in gray); (**C**) in a selective region showing differences between the two HRF models; (**D**) peri-stimulus Time histogram of the same voxel in panel C as a function of HRF type (95% confidence intervals shaded in gray).

**Figure 6 behavsci-15-01457-f006:**
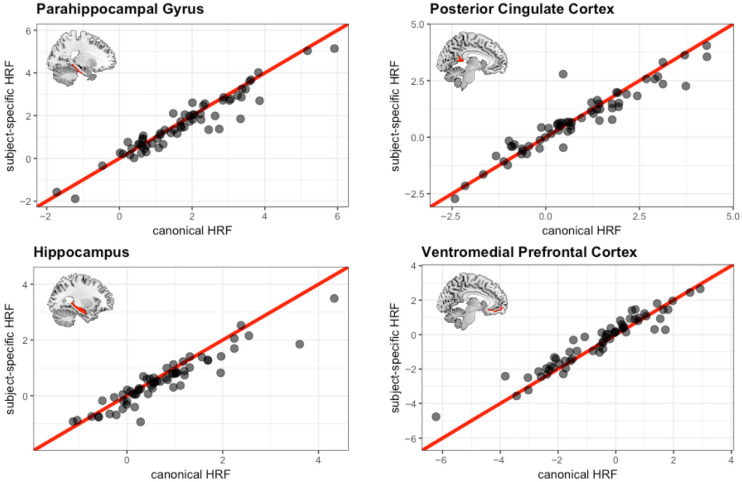
Scatterplots illustrating the association between parameter estimates using the canonical HRF on the *x*-axis and those using the subject-specific HRF on the *y*-axis. The red line represents an identity line (y = x).

**Figure 7 behavsci-15-01457-f007:**
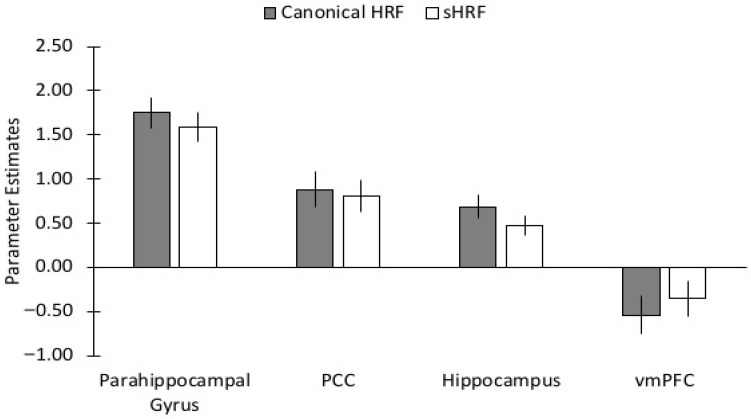
Bar graphs of the parameter estimates comparing the canonical hemodynamic response (gray bars) to the subject-specific HRF (white bars) for the four regions of interest. PCC = posterior cingulate cortex; vmPFC = ventromedial prefrontal cortex.

**Figure 8 behavsci-15-01457-f008:**
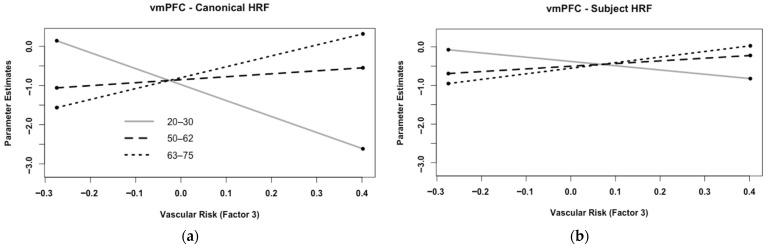
A regression plot of the Age × Vascular Risk interaction for Factor 3 (Diabetic Syndrome Risk) in the vmPFC: (**a**) when using the canonical HRF, as vascular risk increased greater deactivations in the vmPFC were found for younger adults (gray solid line), a smaller relationship was found for middle-aged adults (black dashed lined), and the relationship reversed older adults (black dotted line). Specifically, older adults exhibited less deactivation in the vmPFC as Diabetic Syndrome Risk increased; (**b**) the same pattern was found for the sHRF, but the pattern was muted. vmPFC = ventromedial prefrontal cortex; sHRF = subject-specific HRF.

**Figure 9 behavsci-15-01457-f009:**
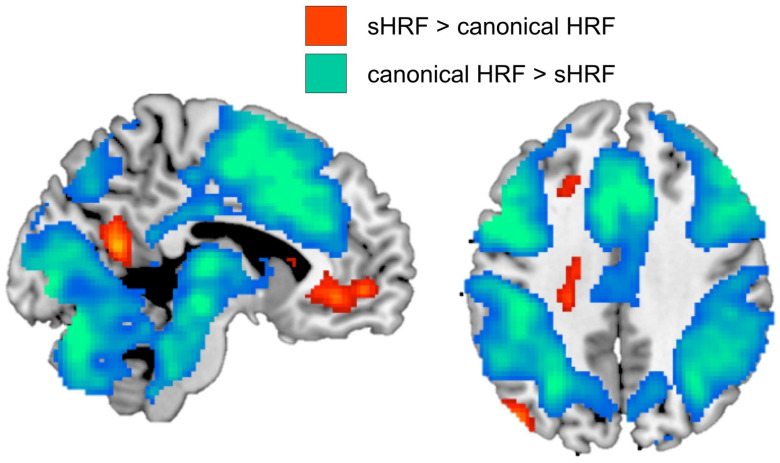
Whole-brain effects for the sHRF compared with the canonical HRF in a paired *t*-test. Using the sHRF for each subject reduced brain activity in task-positive regions and increased brain activity (i.e., reduced deactivations) in task-negative regions.

**Table 1 behavsci-15-01457-t001:** Participant Characteristics in Final Sample.

Factor	Young (20–30)	Middle-Age (50–62)	Old (63–74)
N	19	24	20
Sex (F/M)	10/9	17/7	13/7
Race			
Non-Hispanic White (N/%)	12/63.16%	16/66.67%	17/85%
African American (N/%)	1/5.26%	6/25%	3/15%
Other (N/%)	6/31.58%	2/8.33%	0/0%
Education (M/SD)	15.16/2.14	14.67/2.26	14.10/2.94
SLUMS (M/SD)	NA	27.17/2.96	26.53/2.29
fMRI Memory Accuracy % (M/SD)	56.79/16.96	38.96/14.03	30.90/6.86
Arterial Stiffness (M/SD)	37.05/12.27	46.86/10.11	53.50/12.83
Body Mass Index (M/SD)	25.47/5.61	29.98/7.79	27.76/4.84
Visceral Fat (M/SD)	6.16/4.43	9.92/4.61	11.32/5.63
Abdominal Circumference in cm (M/SD)	90.84/14.38	103.10/22.08	103.42/19.82
Body Fat % (M/SD)	30.19/9.84	39.57/11.44	35.19/10.00
Presence of Diabetes (N/%)	1/5.26%	3/12.5%	6/30%
Family History of Diabetes (N/%) *	2/10.53%	14/58.33%	12/60%
High Cholesterol (N/%) *	0/0%	9/37.5%	13/65%
History of Heart Attack (N/%)	0/0%	3/12.5%	5/25%
Hypertension (N/%) *	0/0%	8/33.33%	12/60%
Smoking Status (N/%)			
Never Smoked	15/78.95%	14/58.33%	12/60%
Quit	2/10.52%	8/33.33%	6/30%
Current	2/10.52%	2/8.33%	2/10%
Gait Speed in ms (M/SD)	2580.95/347.53	2837.29/521.80	3030.26/776.90

Notes: * age differences *p* < 0.05; M = mean, SD = standard deviation.

## Data Availability

The datasets presented in this article are not readily available because the data are part of an ongoing study. Requests to access the datasets should be directed to imcdonough@binghamton.edu.

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
