# Peer review of "Interpreting fMRI Studies in Populations with Cerebrovascular Risk: The Use of a Subject-Specific Hemodynamic Response Function"

_behavsci, 2025, doi:10.3390/bs15111457_

Round 1
Reviewer 1 Report
Comments and Suggestions for Authors
In McDonough et al., the authors tested the effect of health risk factors and age on the validity of using the canonical HRF in modeling task-induced BOLD signals. Using a visual checkerboard task, the authors found that the shape of the hemodynamic response was affected by age and vascular risks. Using a memory encoding task, the authors found overall weaker signal, whether activation or deactivation, when using sHRF versus the canonical HRF. Moreover, activation level modeled with sHRF was less sensitive to age and vascular risk.
The research question is of high importance because it touches the fundamentals and the validity of using fMRI to infer brain state, especially those used with special populations such as aging and brain stroke. However, the major problem to me is that the authors did not provide a sufficient reason for using sHRF instead of the canonical HRF in this case because there is no evidence that the canonical HRF did not fit, hence the conclusions seems unjustified. Comments are listed below:
- Although the research question is of high relevance, from the results I do not see a clear motivation of using sHRF versus the canonical HRF because the shape of the participants’ hemodynamic response did not seem to violate the canonical HRF. One of the major reasons for using customized HRF is when there are delays in a participant’s hemodynamic response such that the canonical HRF would not capture the response peak (Calhoun et al., 2003; Amemiya et al., 2012; Liu et al., 2025). From the checkerboard task, neither the latency nor the FWHM varied with age or health risks, then how is the canonical HRF flawed in this population and why should one use the sHRF?
- The authors should present what the sHRF’s and the canonical HRF look like (i.e. how they differ) and compare it to the time series of participants’ BOLD signal. Visualization such as Fig.2 in Calhoun et al., 2004 and in Figure S4 in Liu et al., 2025 would be necessary to support the advantage of the sHRF over the canonical HRF. Evidence for better model fitting with sHRF versus the canonical HRF is also necessary.
- Are the sHRF and canonical HRF scaled equally? In Figure 6, there is an overall smaller absolute parameter estimate from sHRF than canonical HRF, could it be that the sHRF regressor has a smaller scale than the canonical HRF?
- Usually a better model would yield greater activation, why did the sHRF yield weaker activity? Again, it is unclear in what way it is “better” than the canonical HRF, making the results uninterpretable.
- The division between middle-aged adults and older adults seems rather arbitrary and the effect of age and risk level might vary if a different cutoff was chosen. The authors should present evidence that the results are robust to different cutoffs. Also, why not treat age as a continuous variable?
- How were “Low Risk” and “High Risk” defined for each factor? Were the effect of age and risk level robust to different thresholds of high vs. low risk?
References
Calhoun, V. D., Stevens, M. C., Pearlson, G. D., & Kiehl, K. A. (2004). fMRI analysis with the general linear model: removal of latency-induced amplitude bias by incorporation of hemodynamic derivative terms. Neuroimage, 22(1), 252-257.
Amemiya, S., Kunimatsu, A., Saito, N., & Ohtomo, K. (2012). Impaired hemodynamic response in the ischemic brain assessed with BOLD fMRI. Neuroimage, 61(3), 579-590.
Liu, Y., Halfen, E. J., Yau, J. M., Fischer-Baum, S., Kohler, P. J., Faseyitan, O., ... & Medina, J. (2025). Reweighting of visuomotor areas during motor processing subsequent to somatosensory cortical damage. NeuroImage, 317, 121336.
Author Response
Comments 1: Although the research question is of high relevance, from the results I do not see a clear motivation of using sHRF versus the canonical HRF because the shape of the participants’ hemodynamic response did not seem to violate the canonical HRF. One of the major reasons for using customized HRF is when there are delays in a participant’s hemodynamic response such that the canonical HRF would not capture the response peak (Calhoun et al., 2003; Amemiya et al., 2012; Liu et al., 2025). From the checkerboard task, neither the latency nor the FWHM varied with age or health risks, then how is the canonical HRF flawed in this population and why should one use the sHRF?
Response 1: Thank you for your comments. The motivation for using the sHRF comes from studies that have already shown that relying on the canonical HRF is insufficient to accurately capture neural activity among older adults. We have now added a paragraph in the Introduction to better motivate the study:
Correcting the task-related BOLD signal using cerebrovascular reactivity via a hypercapnic challenge altered the BOLD signal in three ways: 1) uncorrected age-related decreases in the MTL and lingual gyrus were no longer associated with age after correction, 2) uncorrected associations with age now showed significant increases with age in some regions like the left prefrontal cortex, and 3) uncorrected age-related increases in the right prefrontal cortex now revealed highly significant positive associations with age after correction (Liu et al., 2013). Other research has used less-invasive techniques such as correcting the BOLD signal using resting-state fluctuation analyses (RSFA) that have been shown to be correlated with vascular-related physiological functioning rather than neuronal functioning. In one eloquent example, Tsvevtanov et al. (2015) showed that RSFA was related cardiovascular function (e.g., heart rate and heart rate variability) but not neural activity estimated using magnetoencephalography. Furthermore, this study showed that RSFA scaling on task-related BOLD signal resulted in large reductions of age-related differences, suggesting that much of the previous research has overestimated the effects of aging on neural activity when using un-corrected BOLD signals. Liu et al. (2013) also showed that the BOLD signal was related to both cerebrovascular reactivity and RSFA (which were related to each other), providing independent and converging evidence that vascular functioning can modulate the BOLD signal. Together, these studies suggest that the BOLD signal is influenced by cerebrovascular (non-neural) signals in quite unpredictable ways.
The problem is there are not great solutions. The present study attempted to offer a simple solution. However, whether it “works” or not we didn't know from the outset.
Yes delays in response is one common reason, but there is no reason to think that is the only reason. In fact, that logic alone ignores changes in vasculature structure that accompanies aging and poor health.
Perhaps one less well-known reason is amplitude or a combination of other reasons. What we show is that changes in easy to quantify metrics like timing do not accurately capture differences that are found by modeling the HRF itself. In other words, there are qualitative (not quantitative) changes in the shape of the HRF that alters the modeling of the BOLD signal.
Comments 2: The authors should present what the sHRF’s and the canonical HRF look like (i.e. how they differ) and compare it to the time series of participants’ BOLD signal. Visualization such as Fig.2 in Calhoun et al., 2004 and in Figure S4 in Liu et al., 2025 would be necessary to support the advantage of the sHRF over the canonical HRF. Evidence for better model fitting with sHRF versus the canonical HRF is also necessary.
Response 2: Thank you for this suggestion. We looked at those papers and figures and have now incorporated (Figure 5). We show that the sHRF is not universally helpful across all brain regions but can she an improvement in fit similar to the figures shown in the aforementioned articles.
Comments 3: Are the sHRF and canonical HRF scaled equally? In Figure 6, there is an overall smaller absolute parameter estimate from sHRF than canonical HRF, could it be that the sHRF regressor has a smaller scale than the canonical HRF?
Response 3: That is an interesting point we had not considered. While it is possible that the scale is smaller, the figure you mentioned shows that the difference is not dramatically different. We have remade Figure 1 as well to overlay the canonical HRF and we see a similar pattern: The canonical HRF is larger than nearly every sHRF. This difference underscores how the canonical HRF might be overestimating a person’s HRF. Other than scaling per se, it could be the case that the error is smaller in the sHRF thereby becoming more sensitive to individual differences. As argued in the Discussion, it appears that using an sHRF reduced the effects of outlying parameter estimates, thus leading to an overall average decrease but this was region dependent.
Comments 4: Usually a better model would yield greater activation, why did the sHRF yield weaker activity? Again, it is unclear in what way it is “better” than the canonical HRF, making the results uninterpretable.
Response 4: We agree that it many circumstances, greater activity should be found. However, as now elaborated to a greater extent in the Introduction, correcting for cerebrovascular effects on the BOLD signal, in general, have shown both increases in brain activity and decreases. Indeed, in other studies in which we have used this technique, both increases in BOLD signal and decreases have been found consistent with other methods of correction (e.g., McDonough et al., 2019). We argue that the model is a more accurate reflection of aging and health—which might not always be increases in brain activity. We mention this point in the Discussion section. We also show that when estimating the effects of age and vascular risk on brain activity during memory encoding, we found evidence consistent with our claim that variance or error is reduced. Using the sHRF resulted in more consistent and larger effects than when using the canonical HRF. Specifically, the magnitudes of the effects often were larger and the cluster sizes also were larger when using the sHRF compared with the canonical HRF. We agree that “better” is hard to say for certain without a gold standard like actual neural activity to compare against. We now mention this limitation in the Discussion section as well. We also dedicate a large discussion of “Alternative Interpretations” where we lay out the possibility of a poorer fit and a weaker signal. We feel that our study is not the definitive end to the story, but a piece of the puzzle that needs to be followed up on. We also note that most aging fMRI studies offer no methods of BOLD signal correction and at least we offer an easy option that can be conducted alongside the canonical HRF at minimum.
Comments 5: The division between middle-aged adults and older adults seems rather arbitrary and the effect of age and risk level might vary if a different cutoff was chosen. The authors should present evidence that the results are robust to different cutoffs. Also, why not treat age as a continuous variable?
Response 5: We completely agree that she thresholds are arbitrary. We have simply chosen a number to give the reader an idea of the difference in sample subgroups. All analyses used age as a continuous measure age and the split is simply for visualization purposes. This has now been clarified in the Method section.
Comments 6: How were “Low Risk” and “High Risk” defined for each factor? Were the effect of age and risk level robust to different thresholds of high vs. low risk?
Response 6: As now clarified, all analyses used continuous scales. For visual purposes, we used a median split.
Reviewer 2 Report
Comments and Suggestions for Authors
This manuscript addresses an important methodological issue in fMRI research: the influence of vascular risk and aging on the hemodynamic response function (HRF), and whether using subject-specific HRFs (sHRFs) improves interpretation. The study is timely, methodologically careful, and transparent, with clear implications for aging and dementia research. I believe it makes a meaningful contribution, but several aspects require clarification and revision before publication.
The focus on vascular risk factors in shaping HRF variability is highly relevant to neuroimaging in aging and disease populations.
The integration of subject-specific HRFs into analyses offers a useful methodological advance.
The relatively large sample size compared to prior studies in this domain is commendable.
The manuscript offers careful discussion of the limitations of canonical HRF assumptions, which will be valuable to the field.
Points for Revision
- The sHRF was derived from a single voxel in the occipital cortex and then applied across the brain. This assumption of global HRF generalizability is not clearly justified and may not capture region-specific variability. Please provide stronger rationale for this choice, and consider discussing how results might differ if HRFs were estimated regionally. Instead of using just one region, how do sHRF vary for one individual across the brain, e.g. for auditory, motor, sensorimotor areas.
- Although larger than some earlier HRF variability studies, the sample (n = 63) remains relatively small for disentangling complex age × vascular risk interactions. Please temper generalizations about broad population effects and highlight this limitation more explicitly.
- The differences between canonical HRF and sHRF analyses, while statistically significant in some cases, often appear modest. The manuscript at times implies stronger benefits of sHRFs than the data fully support. I encourage you to more clearly delimit the claims to emphasize that findings are preliminary rather than conclusive evidence of superiority. However, the increase of T-values when using sHRF might point towards a better fit of the underlying GLM model for individual subjects than for using the canonical HRF. Maybe, canonical HRF overestimates underlying neural activity?
- Figures 2–4 are informative but could be made clearer with more consistent labeling and color schemes for age groups and risk categories. In figure 2 the authors claim to illustrate differences in sHRF for low vs. high risk for younger and older subjects, however this is not clearly visible from the graphs. Also, there is a second figure 2 on page 16 which shows more clearly the differences in sHRF mentioned before, however, the caption does not fit to the first figure 2 (Factor 3 vs. Factor 1). Also, in all three figures, 2 to 4, there is the same typo: Olde instead Older? Please check for typographical errors and formatting inconsistencies, particularly in figure captions.
- The authors mention to have used MPRAGE whereas they mention two TI times. Did they indeed use MPRAGE or more likely MP2RAGE?
This manuscript could impact how fMRI studies of aging and vascular risk are conducted and interpreted.
Author Response
Comments 1: The sHRF was derived from a single voxel in the occipital cortex and then applied across the brain. This assumption of global HRF generalizability is not clearly justified and may not capture region-specific variability. Please provide stronger rationale for this choice, and consider discussing how results might differ if HRFs were estimated regionally. Instead of using just one region, how do sHRF vary for one individual across the brain, e.g. for auditory, motor, sensorimotor areas.
Response 1: We completely agree that we are making a large assumption through this technique. However, we justify it through prior research that has shown that inter-individual differences in the HRF are much greater than intra-individual differences (Handwerker et al., 2004; Miezin et al., 2000). Nonetheless, we agree that we need to test its effectiveness or lack therefore of rather than assuming it is an ineffective strategy. We chose a checkerboard task because we, as a science, have a very good understanding of what stimulates this area of cortex. Ideally, we would have a task for each voxel. Of the brain but that would not be feasible. Furthermore, we have a much poorer understanding of what maximally stimulates must other regions of the brain and so we would not know what task to even choose for many other regions (hence the point of much fmri research). Thus, this point balances practically and feasibility. We discuss these ideas in section 4.1 of the Discussion.
Comments 2: Although larger than some earlier HRF variability studies, the sample (n = 63) remains relatively small for disentangling complex age × vascular risk interactions. Please temper generalizations about broad population effects and highlight this limitation more explicitly.
Response 2: Thank you for this comment. We now explicitly note in the Discussion (Section 4.2) that our sample size is small and our goal is to show a proof-of-concept rather than to make large generalizable claims. We have also tempered our wording in the Conclusion.
Comments 3: The differences between canonical HRF and sHRF analyses, while statistically significant in some cases, often appear modest. The manuscript at times implies stronger benefits of sHRFs than the data fully support. I encourage you to more clearly delimit the claims to emphasize that findings are preliminary rather than conclusive evidence of superiority. However, the increase of T-values when using sHRF might point towards a better fit of the underlying GLM model for individual subjects than for using the canonical HRF. Maybe, canonical HRF overestimates underlying neural activity?
Response 3: Thank you for this comment. In addition to tempering our wording as mentioned in the last point, we mention several times in the Discussion that more work needs to be done to validate the use of an sHRF. We do also discuss the possibility of the canonical HRF overestimating neural activity but again, more work is needed, especially with a “gold standard” of neural activity, which we did not have available.
Comments 4: Figures 2–4 are informative but could be made clearer with more consistent labeling and color schemes for age groups and risk categories. In figure 2 the authors claim to illustrate differences in sHRF for low vs. high risk for younger and older subjects, however this is not clearly visible from the graphs. Also, there is a second figure 2 on page 16 which shows more clearly the differences in sHRF mentioned before, however, the caption does not fit to the first figure 2 (Factor 3 vs. Factor 1). Also, in all three figures, 2 to 4, there is the same typo: Olde instead Older? Please check for typographical errors and formatting inconsistencies, particularly in figure captions.
Response 4: Thank you for pointing out these mistakes, which have now been corrected. We also have now elaborated in the text how to interpret the patterns shown in the figures to improve their clarification.
Comments 5: The authors mention to have used MPRAGE whereas they mention two TI times. Did they indeed use MPRAGE or more likely MP2RAGE?
Response 5: That is correct. We have corrected this typo.
Round 2
Reviewer 1 Report
Comments and Suggestions for Authors
I appreciate the changes made to the text and the figures. The Introduction is improved by the added contents. In my opinion, what is still missing is the necessity and benefit of using sHRF over the canonical HRF. If all what was influenced by age and risk factor was just response amplitude instead of the shape of the response per se, then it can be well captured by the canonical HRF and controlled with age and risk factor as covariates of no interest. The authors argued that using sHRF may have reduced noise without providing evidence. Evaluation of model fitting would help strengthen the point.
Also, although I suggested Figure 5, it may not make a lot sense displaying the whole timeseries from an event-related design because the alignment between the models and the BOLD response is difficult to assess. I'd make it an event-related averaging (i.e. peristimulus plot) for a better comparison.
Author Response
Comments 1: I appreciate the changes made to the text and the figures. The Introduction is improved by the added contents.
Response 1: Thank you.
Comments 2: In my opinion, what is still missing is the necessity and benefit of using sHRF over the canonical HRF. If all what was influenced by age and risk factor was just response amplitude instead of the shape of the response per se, then it can be well captured by the canonical HRF and controlled with age and risk factor as covariates of no interest. The authors argued that using sHRF may have reduced noise without providing evidence. Evaluation of model fitting would help strengthen the point.
Response 2: If I understand correctly, the argument is that controlling for age and vascular risk should be sufficient to account for differences in the hemodynamic response and implementing the sHRF does not show anything new or different. However, we have shown that the sHRF does reveal different patterns than the canonical HRF even when age and vascular risk are controlled in both models:
- The interaction between Diabetic Syndrome Risk and brain activity in the parahippocampal gyrus and vmPFC are significant using the canonical HRF but not in the sHRF and these differences were significant. Although we acknowledge in the paper that there are multiple interpretations for this finding, one implication is that the canonical HRF over-estimates relationships with vascular risk. Regardless of the interpretation, the inference would be very different if using the canonical HRF versus the sHRF.
- The effect of age on encoding-related brain activity when controlling for Factor 1 effects revealed many similarities but also some qualitative differences such as significance in the left caudate using the canonical HRF (which was absent in the sHRF) and significance in the left cingulate cortex using the sHRF (which was absent in the canonical HRF); only left middle temporal gyrus showed a significant age x Factor 1 risk when using the canonical HRF but the left middle temporal gyrus, left caudate, and the left cerebellum were significant when using the sHRF. There are other examples as well (this was not the only one). We realized these differences may have not been obvious and so we added figures in the supplementary material to highlight these differences.
- For the brain-behavior correlations, the relationship between parahippocampal gyrus and associative memory accuracy was only significant when using the sHRF and not the canonical HRF. Accordingly, the error in the model also is numerically lower (Residual standard error for canonical = 16.45, and for sHRF = 16.26)
Thus, what is the necessity and benefit of using the sHRF over the canonical HRF? It appears that one’s inferences can qualitatively change such that using the sHRF reveals new patterns of relationships not otherwise found by the canonical HRF.
Comments 3: Also, although I suggested Figure 5, it may not make a lot sense displaying the whole timeseries from an event-related design because the alignment between the models and the BOLD response is difficult to assess. I'd make it an event-related averaging (i.e. peristimulus plot) for a better comparison.
Response 3: We agree that the original suggested figure style fits better with a blocked design, but still feel that it has some merit when at least comparing canonical and sHRF adjusted models, even with the shift relative to the raw time series. Thus, we have kept the original figures and added the PSTH for the same voxels as the original time series also in Figure 5. We are not convinced that this new addition reveals anything qualitatively different but readers can decide for themselves how helpful each figure is.
Reviewer 2 Report
Comments and Suggestions for Authors
I have no further comments to add.
Author Response
Comments 1: I have no further comments to add.
Response 1: Thank you.